# *Mlig-SKP1* Gene Is Required for Spermatogenesis in the Flatworm *Macrostomum lignano*

**DOI:** 10.3390/ijms232315110

**Published:** 2022-12-01

**Authors:** Mikhail Biryukov, Anastasia Dmitrieva, Valeriya Vavilova, Kirill Ustyantsev, Erzhena Bazarova, Igor Sukhikh, Eugene Berezikov, Alexandr Blinov

**Affiliations:** 1Institute of Cytology and Genetics, Siberian Branch of Russian Academy of Science, 630090 Novosibirsk, Russia; 2European Research Institute for the Biology of Ageing, University of Groningen, University Medical Center Groningen, 9700AD Groningen, The Netherlands

**Keywords:** spermatogenesis, regeneration, proliferation, stem cells, flatworms, RNA interference

## Abstract

In a free-living flatworm, *Macrostomum lignano*, an S-phase kinase-associated protein 1 (*SKP1*) homologous gene was identified as enriched in proliferating cells, suggesting that it can function in the regulation of stem cells or germline cells since these are the only two types of proliferating cells in flatworms. *SKP1* is a conserved protein that plays a role in ubiquitination processes as a part of the Skp1-Cullin 1-F-box (SCF) ubiquitin ligase complex. However, the exact role of *Mlig-SKP1* in *M. lignano* was not established. Here, we demonstrate that *Mlig-SKP1* is neither involved in stem cell regulation during homeostasis, nor in regeneration, but is required for spermatogenesis. *Mlig-SKP1*(RNAi) animals have increased testes size and decreased fertility as a result of the aberrant maturation of sperm cells. Our findings reinforce the role of ubiquitination pathways in germ cell regulation and demonstrate the conserved role of *SKP1* in spermatogenesis.

## 1. Introduction

*Macrostomum lignano* is a free-living marine flatworm that in recent years has been developed into a potent model organism to study regeneration and a multitude of other biological phenomena [1]. *M. lignano* is easy to culture in a laboratory, it has a short generation time, small size, and transparent body, and can regenerate most tissues and organs. It is a hermaphroditic organism that reproduces exclusively in a sexual manner [2]. Numerous experimental methods have been developed to study *M. lignano* biology, including antibody labeling, in situ hybridization (ISH), RNA interference (RNAi), and transgenesis [3,4,5,6,7,8,9,10,11].

The regeneration of *M. lignano*, which is based on the proliferation of somatic stem cells (neoblasts), is one of the major focus areas for research using this model organism [1,12,13]. It was shown that in *M. lignano*, neoblasts could differentiate into almost any cell type within several days [14]. Furthermore, substantial information about molecular markers of proliferating cells in *M. lignano* has been generated [3,7,8,12,15,16], including transcriptional signatures of neoblasts and germline cells [12]. However, for a deeper understanding of the molecular mechanisms that regulate the regeneration and maintenance of neoblasts and germline cells, functional characterization of the genes specifically expressed in these cells is required. In this work, we focus on one such gene, a homolog of the human *SKP1* gene, which was previously identified as specifically expressed in proliferating cells of *M. lignano* [12]. 

S-phase kinase-associated protein 1 (*SKP1*) is a core component of the SCF (Skp1-Cullin 1-F-box) ubiquitin ligase complex [17]. Ubiquitination is an intrinsic process of protein degradation, which occurs in all cells of the organism, serving to ensure cell homeostasis [18]. In a nematode, *Caenorhabditis elegans*, it was shown that *SKP1*-related genes perform critical functions in regulating cell proliferation, meiosis, and morphogenesis [19]. In mammals, *SKP1* is involved in the regulation of double-stranded DNA breaks and spermatogenesis [20,21]. In a planarian, *Schmidtea mediterranea*, knockdown of the *SKP1* gene leads to regenerative defects, such as the impaired formation of blastema and the brain [22]. In *M. lignano*, an *SKP1* homolog (*Mlig-SKP1*) is also expressed in proliferating cells, but its functions have not been demonstrated. Therefore, we investigated the effects of *Mlig-SKP1* mRNA knockdown on *M. lignano* using RNA interference (RNAi) in order to determine the role of the gene in regeneration and, consequently, its role in neoblast activity. Unexpectedly, we found that the knockdown has no noticeable effect on neoblast activity, but it has a great impact on the animal’s fertility, rendering it sterile in certain conditions. Our results show that the *Mlig-SKP1* gene plays a major role in spermatogenesis in *M. lignano*.

## 2. Results and Discussion

### 2.1. Knockdown of Mlig-SKP1 Gene Does Not Affect Regeneration but Causes Abnormal Testes Morphology

An *M. lignano* gene homologous to human and mouse *SKP1*, hereafter *Mlig-SKP1* (Mlig014611.g1), was selected from a previously established list of markers for proliferating cells [12]. We used RNAi to knockdown expression of *Mlig-SKP1* in homeostatic and regenerative conditions. A high efficiency of gene knockdown by soaking animals in *Mlig-SKP1* dsRNA-containing medium was confirmed by qPCR (Appendix A). During homeostasis, knockdown of *Mlig-SKP1* led to an obvious phenotype, which manifested in an increased size of the worms’ testes (Figure 1). After the first two weeks of RNAi, the enlarged testes phenotype was observed in up to 70% of treated animals. However, with the prolonged dsRNA treatment, some animals restored the size of their testes, and after 3–5 days of RNAi, the phenotype penetrance varied between 5% and 38% (median 25%, Appendix A) and animals with a varying degree of altered testes size could be observed (Appendix A). 

Knockdown of genes important for neoblast activity, such as, for example, *Mlig-DDX39* and *Mlig-PIWI* [12], usually results in obvious phenotypes in *M. lignano*, leading to the loss of tissue homeostasis, the inability to regenerate, and, eventually, lethality. In the case of *Mlig-SKP1*, we did not observe such phenotypes in homeostatic conditions, and enlarged testes was the only discernible phenotype, suggesting that *Mlig-SKP1* does not play a major role in neoblasts’ maintenance. To find out a role that *Mlig-SKP1* could play in regeneration, we performed additional RNAi experiments, where animals pretreated with dsRNA against *Mlig-SKP1* were amputated and allowed to regenerate. However, we did not observed any difference in regeneration between the treated and control animals.

### 2.2. Mlig-SKP1 Is Expressed in Gonads

Knowing the expression pattern of *Mlig-SKP1* should help to unveil the cells, organs, and tissues that could be affected by RNAi. We performed whole-mount in situ hybridization to track where the gene is expressed in adult animals. It appears that *Mlig-SKP1* is primarily expressed in the testes and to a less extent in the ovaries (Figure 2). All other weakly stained parts of the pattern vary between animals and likely represent unspecific staining. Importantly, the expression of *Mlig-SKP1* is localized to the outer part of the testes, or testes periphery, where spermatogonia and spermatocytes reside and proliferation takes place [15,23].

### 2.3. Mlig-SKP1 Knockdown Leads to Decreased Fertility

To investigate whether the changes in testes morphology in *Mlig-SKP1*(RNAi) animals would affect their fertility, we performed fertility measurement experiments. For this, adult animals were treated with dsRNA against *Mlig-SKP1* for two weeks and then were put together in new experimental wells to cross-fertilize and lay eggs for four days, after which the animals were removed from the wells, and the laid eggs were allowed to develop. One week later, the number of hatchlings in each well was counted and used to assess the fertility of the worms. While control *GFP* (RNAi) animals on average produced 0.9625 ± 0.2054 progeny per parent, *Mlig-SKP1*(RNAi) animals produced 0.3125 ± 0.2126 progeny per parent (Appendix A) (*p*-value < 0.001 using *t*-test).

To investigate whether gonad development is affected in *Mlig-SKP1*(RNAi) animals, we performed additional RNAi experiments in 3–5-day-old hatchlings. We observed only a slight delay of 2–3 days in the maturation of *Mlig-SKP1*(RNAi) hatchlings, but otherwise the fertility of these young adult animals was the same as for the animals where knockdown was started only in the adult stage. Interestingly, after the dsRNA treatment was stopped, the fertility in both groups of animals recovered in a period of 1–2 weeks. These observations suggest that *Mlig-SKP1* does not play a role in the development of tissues other than gonads, and even there its absence shows only temporal changes.

### 2.4. Mlig-SKP1 Knockdown Results in Aberrant Morphology of Sperm Cells

To further characterize changes in the testes of *Mlig-SKP1*(RNAi) animals, we performed microscopy analysis, which revealed that the increased volume of the testes might be a result of an accumulation of aberrant sperm cells. The sperm in *Mlig-SKP1*(RNAi) animals looks disorganized and sphere-like (Figure 3), suggesting that either the development was stopped at one of the maturation steps or the dysregulation led to the wrong sperm shape. The form of immature sperm partly resembles *Mlig-elav* [24] and *Mlig-sperm1* [15] knockdown phenotypes. In contrast to the *Mlig-elav* knockdown, *Mlig-SKP1*(RNAi) sperm cells have only one short tail of spermatids instead of four, and in comparison to the *Mlig-sperm1* knockdown, the same disorders (wrong shape of the flagellum and head) seem to be involved.

### 2.5. Conserved Role of SKP1 in Regulation of Spermatogenesis

The *Mlig-SKP1* gene characterized in this work shows no connection to the regulation of neoblasts. Instead, we found that it plays a role in the spermatogenesis process. The lack of *Mlig-SKP1* results in the disruption of sperm maturation, leading to the increased size of testes and decreased fertility.

According to previous studies of *SKP1*, the main role of the *SKP1* protein in mammals is being a part of a complex responsible for ubiquitination [17,18]. Importantly, *SKP1* was also recently demonstrated to play roles in early meiotic processes in mammals [21], and in spermatogenesis in particular [20], by regulating DNA double-stranded break pathways. Hence, our demonstration that in a flatworm *SKP1* is also required for spermatogenesis indicates that this is a deeply conserved function of this gene.

## 3. Materials and Methods

### 3.1. Organism and Its Culture Conditions

*Macrostomum lignano* [1] flatworms were kept in glass Petri dishes with nutrient-enriched artificial seawater (Guillard’s f/2 medium [25]), at 20 °C and in a 14:10 hours light/dark day cycle and fed with the diatom *Nitzschia curvilineata* ad libitum. In the present study, we used the wild-type NL12 strain [9].

### 3.2. RNA Interference

RNAi treatment of *M. lignano* worms was performed by soaking as previously described [12,26,27]. We used 20–25 animals per well of a 24-well plate for *GFP* as a control and for *Mlig-SKP1* RNAi treatment, respectively. Twenty-day-old worms were maintained in 24-well plates and incubated with 300 μL dsRNA solution (10.0 ng/μL) in f/2 medium containing diatoms. The dsRNA solution was changed daily. In a part of experiments, half of the wells for both *Mlig-SKP1* knock down and control worms were amputated above testes after 2 weeks of RNAi treatment. After 3–4 weeks of RNAi treatment, when all amputated worms were recovered, the specimens were examined with Zeiss Axio Zoom V16 stereomicroscope equipped with an HRm digital camera (Zeiss, Jena, Germany). For the observation of testis components and sperm morphology, live worms were taken randomly and relaxed in MgCl_2_ 7.14% of f/2 medium. Then the testis were directly observed whole and after the testis ruptured using Zeiss LSM 780 inverted microscope equipped with DIC optics.

### 3.3. Whole-Mount In Situ Hybridization

The colorimetric whole-mount in situ hybridization (ISH) with a PCR-amplified DIG-labeled antisense RNA probe was performed according to previously described protocols [1,15,27,28]. The synthesis of cDNA was performed using the Reverse transcriptase RNAscribe RT (Biolabmix, Novosibirsk, Russia) according to the manufacturer’s protocol. Oligo(dT) and random hexamer primers were used with a total RNA amount about 2–3 µg as a template for each reaction. DNA fragments of *Mlig-SKP1* were amplified from cDNA by standard PCR with BioMaster LR HS-Taq PCR (2×), followed by cloning into the pGEM-T-Easy vector system (Promega, Madison, WI, USA) and sequenced by the SB RAS Genomics Core Facility (Novosibirsk, Russia, http://sequest.niboch.nsc.ru). Primers are listed in Appendix A. DNA templates containing T7 promoter sequences at the 5′ ends of both strands for producing DIG-labelled riboprobes were amplified from the sequenced plasmid using High Fidelity Pfu polymerase (Thermo Scientific, Waltham, MA, USA). Digoxygenin-labeled RNA probe was generated using DIG RNA Labeling KIT SP6/T7 (Roche, Basel, Switzerland), following the manufacturer’s protocol. The staining was developed using NBT/BCIP colorimetric substrate (Roche). Images were made using a standard Zeiss Axio Scope.A1 inverted microscope equipped with an Axiocam 512 color (Zeiss, Jena, Germany) digital camera.

### 3.4. RNAi Fertility Experiments

To evaluate the effect of *Mlig-SKP1* RNAi knockdown on worms’ fertility, the following experiment was conducted: groups of 20 adult worms were placed in a 24-well plate and daily treated with dsRNA, dissolved in freshly prepared f/2 medium, for three weeks. For fertility measurement, the number of hatchlings in each well was counted twice a week. Moreover, 2–3-day-old hatchlings were taken to start a new experiment with the same conditions.

### 3.5. qRT-PCR for Knock Down Verification

To confirm the efficiency of the RNAi knockdown, we performed qRT-PCR using 3 replicas of 20 worms, which were treated by either *Mlig-SKP1* or *GFP* dsRNA for 2 weeks. After that, each replica was rinsed with a fresh f/2 medium, suspended in 500 μL of TRIzol reagent (Ambion, Austin, TX, USA), and stored at −80 °C. Total RNA was extracted from the samples following the manufacturer’s protocol. The RNA was reverse transcribed to generate cDNA using Reverse transcriptase RNAscribe RT (Biolabmix) with a mix of oligo(dT) and Random Hexamer primers. Expression of *Mlig-SKP1* mRNA was checked along with that of 3 housekeeping genes used in previous studies [15,27]. The primers used are listed in Appendix A. qPCR was performed with the BioMaster UDG HS-qPCR Lo-ROX SYBR (2×) (Biolabmix) using the Light Cycler 96 system (Roche Life Sciences, Basel, Switzerland). Gene expression analysis and statistics were performed with Light Cycler 96 software.

## 4. Conclusions

We investigated the functional role of an *M. lignano* gene homologous to human *SKP1* (*Mlig-SKP1*). After the knockdown of *Mlig-SKP1* by RNAi we observed a phenotype associated with dysfunction of the spermatogenesis process and formation of aberrant sperm, which caused the enlargement of testes and decreased fertility. Recently, a crucial role of *SKP1* in mammalian spermatogenesis was demonstrated [18,20,21], and here, by showing that *SKP1* is required for spermatogenesis in flatworms, we demonstrate that this is an ancient conserved function of this gene.

## Figures and Tables

**Figure 1 ijms-23-15110-f001:**
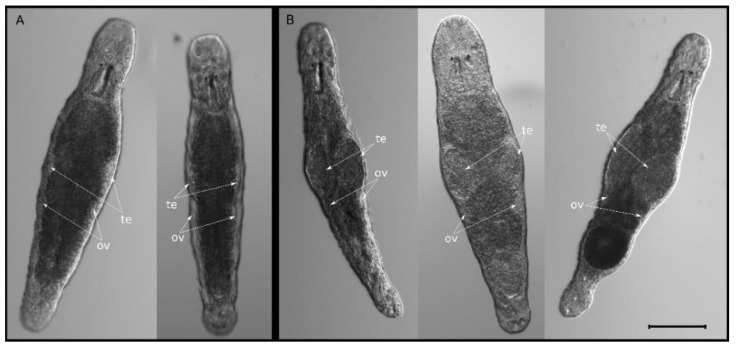
*Mlig-SKP1* RNAi leads to enlarged testes in *M. lignano*. Live worm images are shown. (**A**) Control animals treated with dsRNA against *GFP*. (**B**) Animals treated with dsRNA against *Mlig-SKP1*. Scale bar is 150 µm. Te—testes, ov—ovaries.

**Figure 2 ijms-23-15110-f002:**
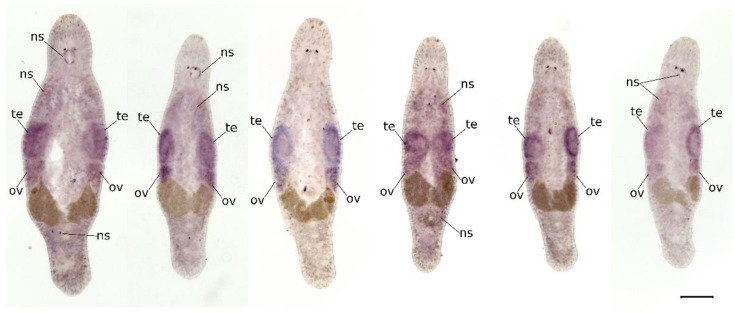
Expression pattern of *Mlig-SKP1* mRNA obtained by whole-mount in situ hybridization performed on homeostatic animals. Expression of *Mlig-SKP1* is observed in testes (te) and to a lesser extent in ovaries (ov). Gray mass below testes and ovaries are developing eggs. Scale bar is 100 µm. ns—non-specific staining.

**Figure 3 ijms-23-15110-f003:**
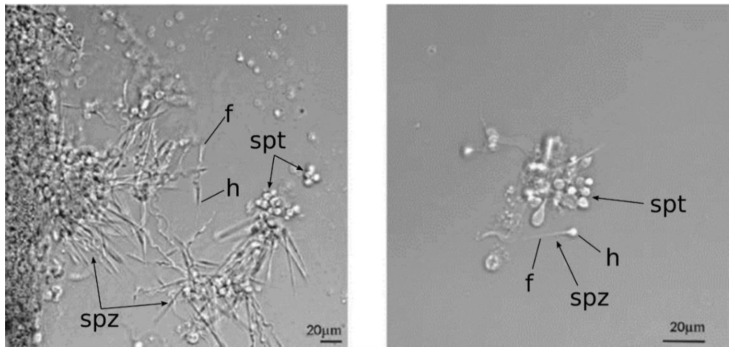
Sperm morphology is affected in *Mlig-SKP1*(RNAi) animals. DIC images are shown. Normal sperm from the worms treated by *GFP* dsRNA is on the left and sperm of the *Mlig-SKP1*(RNAi) worms is on the right. Instead of normal sperm maturation involving splitting of sphere-like tetramers into filamentous mature sperm cells [15], sperm of *Mlig-SKP1*(RNAi) worms is still carrying a sphere-like head and the tail is not well formed. Spermatozoa (spz) and spermatids (spt) are indicated by arrows. h—head of sperm cell, f—flagellum.

## Data Availability

Not applicable.

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
