# Peer review of "Mlig-SKP1* Gene Is Required for Spermatogenesis in the Flatworm *Macrostomum lignano"

_ijms, 2022, doi:10.3390/ijms232315110_

Round 1

Reviewer 1 Report

Abstract: Second sentence can be the first and first can be second. Just a suggestion.

Introduction: Here it is not clear whether the gene's role in spermatogenesis is novel or not. Later from result and conclusion I understood that it is not a novel function. But you are reported first time in flatworm. You can emphasize more on that.

Results and discussion: In Figure 2, "te" and "ov" are not very visible. 

Conclusion: No comments.

Author Response

Dear reviewer,

Thank you for taking the time to review our manuscript and your comments.

We changed our manuscript according to most of your reccommendations. Hereafter, point-by-point:

Point 1: Abstract: Second sentence can be the first and first can be second. Just a suggestion.

Response 1: We appreciate this recommendation and have replaced those two sentences.

Point 2: Introduction: Here it is not clear whether the gene's role in spermatogenesis is novel or not. Later from result and conclusion I understood that it is not a novel function. But you are reported first time in flatworm. You can emphasize more on that. 

Response 2: That is an excellent suggestion, and, indeed, in out introduction in lines 54-56 we have mentioned that: In M. lignano, an SKP1 homolog (Mlig-SKP1) is also expressed in proliferating cells, but its functions have not been demonstrated.

Point 3: Results and discussion: In Figure 2, "te" and "ov" are not very visible. 

Response 3: We updated the Figure 2 according to your racommendation

Again, thanks for your comments

Reviewer 2 Report

The article clearly indicates SKP1 as an essential protein in the processes of spermatogenesis.
The Macrostomum lignano was used as a model to prove this.
The applied research methods fully fit into the experiment.

Author Response

Dear reviewer,

We would like to thank you for taking the time to review our manuscript and for your comments